# The Challenges of Treating Glucokinase MODY during Pregnancy: A Review of Maternal and Fetal Outcomes

**DOI:** 10.3390/ijerph19105980

**Published:** 2022-05-14

**Authors:** Alena Kirzhner, Oren Barak, Edi Vaisbuch, Taiba Zornitzki, Tal Schiller

**Affiliations:** 1Diabetes, Endocrinology and Metabolic Disease Institute, Kaplan Medical Center, Faculty of Medicine, Hebrew University of Jerusalem, Rehovot 9190401, Israel; lior_zo@inter.net.il (T.Z.); schillerta@gmail.com (T.S.); 2Department of Medicine, Kaplan Medical Center, Faculty of Medicine, Hebrew University of Jerusalem, Rehovot 9190401, Israel; 3Department of Obstetrics and Gynecology, Kaplan Medical Center, Faculty of Medicine, Hebrew University of Jerusalem, Rehovot 9190401, Israel; barak.oren@gmail.com (O.B.); ediva1@clalit.org.il (E.V.); 4Magee-Womens Research Institute, University of Pittsburgh School of Medicine, Pittsburgh, PA 15261, USA; 5Department of Obstetrics, Gynecology and Reproductive Science, University of Pittsburgh School of Medicine, Pittsburgh, PA 15261, USA

**Keywords:** diabetes, glucokinase, MODY, MODY 2, pregnancy

## Abstract

Background: The optimal treatment strategy for the follow-up and management of women with glucokinase maturity-onset diabetes of the young (GCK−MODY)during pregnancy remains unknown. Data regarding maternal and fetal outcomes are lacking. Aim: This paper summarizes the existing literature regarding the maternal and fetal outcomes of women with glucokinase MODY to guide future treatment strategy. Methods: A literature search was conducted in Pubmed, Embace, and Cochrane library with citation follow-up using the terms: glucokinase, MODY, diabetes, pregnancy, gestation, and outcomes. We searched for articles with known fetal mutational status. Relevant outcomes included: birthweight, large for gestational age (LGA), small for gestational age (SGA), macrosomia, cesarean delivery (CD), shoulder dystocia, congenital anomalies, miscarriages, preterm births, and long-term outcomes. Results: Fourteen relevant manuscripts were identified describing maternal and fetal outcomes. The percentage of LGA and macrosomia in 102 glucokinase -unaffected offspring (GCK−) was significantly higher than in the glucokinase -affected offspring (GCK+) (44% vs. 10%, *p* < 0.001 and 22% vs. 2%, *p* < 0.001, respectively). Among the 173 GCK(+) offspring, only 5% were SGA, which can be expected according to the normal distribution. We observed higher rates of CD and shoulder dystocia in the GCK(−) offspring. Conclusions: GCK(−) offspring have significantly higher birthweights and more birth complications. The optimal treatment strategy to guide management should take into consideration multiple variables other than fetal mutational status.

## 1. Introduction

Monogenic diabetes accounts for 1–2% of all diabetes cases and is frequently misdiagnosed as type 1, type 2, or gestational diabetes mellitus (DM) [1]. Glucokinase maturity-onset diabetes of the young (GCK−MODY) is an autosomal dominant inherited diabetes caused by mutations in the glucokinase gene [2]. Genetic variants reduce the function of glucokinase, the enzyme in pancreatic beta cells that is the glucose sensor responsible for glucose entry to the glycolytic pathway. This results in reduced sensitivity to glucose-induced insulin secretion and an upward shift in fasting and postprandial blood glucose [3,4].

GCK−MODY is more likely in young (<25 years old), non-obese individuals with mild fasting hyperglycemia on several occasions [5], and a high index of suspicion is needed for correct diagnosis [1]. The guidelines for obtaining a genetic diagnosis [6] include 1. mild fasting hyperglycemia in the range of 99–144 mg/dL that is persistent and stable over several years; 2. HbA1C above the upper limit of the normal range but not above 7.5%; 3. a 75 g oral glucose tolerance test (OGTT) with an increment of blood glucose at two hours (70% of patients tested had an increment less than 54 mg/dL and rarely exceeding 82 mg/dL); and 4. a family history of diabetes. Treatment of GCK−MODY with oral hypoglycemics or insulin does not significantly change glycemic control due to the rapid onset of counter-regulatory mechanisms maintaining glucose concentrations at a higher level [7]. Therefore, treatment, unless during pregnancy, is not recommended [2]. 

The estimated prevalence of GCK−MODY among patients diagnosed with gestational DM (GDM) is 0.4–1%, making this a relatively rare condition during pregnancy [7]. The lack of readily available genetic testing with considerable cost leaves the majority of women undiagnosed before and during pregnancy, leading to very limited clinical experience. The small number of diagnosed women makes outcomes difficult to ascertain, as data are based solely on retrospective cohorts. Randomized controlled trials are virtually impossible to conduct. Recently, it has been suggested that, unlike other types of pre-gestational DM, not all GCK−MODY women should be treated during pregnancy and that treatment initiation should depend on fetal mutational status [7]. The proposed paradigm suggests that if the fetus is a carrier of the GCK maternal mutation GCK(+), no treatment is necessary; however, if the fetus does not carry the mutation GCK(−), glucose-lowering treatment is indicated. In most cases, the fetal mutational status is unknown. Fetal abdominal circumference (AC) growth can then serve as a surrogate marker, and if accelerated (>75th percentile), a non-carrier fetus is suggested, and treatment is indicated [8]. This approach aims to circumvent the unknown mutational status and to reconcile GCK(−) fetuses that need treatment versus GCK(+) fetuses in which unnecessary treatment may potentially increase the risk of small-for-gestational-age (SGA) neonates and its related complications [9]. However, data to support this strategy are limited and mostly rely on case reports and observational cohort studies, some of them published two decades ago with different diagnostic cutoffs and treatments [10,11]. Furthermore, little data exist on miscarriage risk, congenital anomalies, and the long-term consequences of newborns to GCK−MODY mothers to allow informed treatment decisions [12]. 

Thus, the aim of this report is to summarize the existing literature regarding the maternal and fetal outcomes of women with glucokinase MODY to highlight gaps in knowledge in order to guide future treatment strategy. 

## 2. Methods 

A literature search was conducted through MEDLINE, EMBASE, and Cochrane library using the following terms and keywords: glucokinase, MODY, MODY 2, diabetes in pregnancy, gestation, and outcomes. We also searched the citation list of the retrieved articles. All relevant papers until June 2021 were reviewed by two authors. We included papers describing fetal mutational status by genetic analysis describing treatment decisions in GCK(+) versus GCK(−) fetuses and fetal and maternal outcomes. Relevant outcomes included: birthweight, large for gestational age (LGA) > 90th percentile, SGA < 10th percentile, macrosomia > 4 kg, cesarean delivery (CD), shoulder dystocia, congenital anomalies, miscarriages, and preterm births. We also searched for papers describing the long-term outcomes of children born to GCK−MODY-affected women. Our own experience with two women was added. 

### Statistical Analysis

Data are presented as means ± standard deviations or as percentages. The chi-square test was used to assess the relationship between two categorical variables. A *p* value of <0.05 was considered statistically significant. Data were analyzed using SPSS 25 (SPSS Inc., Chicago, IL, USA). 

## 3. Results

Eighty-six relevant manuscripts were retrieved describing GCK−MODY during pregnancy. Fourteen met the search inclusion criteria of mutational status, treatment, and outcomes, including nine retrospective cohorts and five case reports (no randomized controlled trials (RCT’s) were identified). A list of the relevant publications is presented in Table 1. Of note, there is some overlap in the women included in the study by Tinoco et al. [13] that were already published by Bacon et al. [14].

### 3.1. Influence of Insulin Treatment versus Diet on Birthweight and Birth Centile

As seen in Table 2, six reports contain data on the influence of insulin treatment versus diet on birthweight in GCK(−) and GCK(+) offspring, and four of them also specify birth centiles. For the manuscript by Bacon et al. [14] we calculated birth centiles based on the reported birthweight and gestational age at delivery using the fetal growth calculator by the World Health Organization [22]. Birthweight did not differ significantly between insulin-treated and diet-treated GCK(−) offspring in all the cohorts listed. In the GCK(+) offspring, two out of five studies showed significantly larger newborns when diet was given rather than insulin [2,13]. The other three studies did not find significant differences in birthweights. When considering birth centiles, a borderline significance was noticed by Hosokawa et al. [10] in GCK (−) newborns to insulin-treated mothers that were significantly smaller compared to diet-treated mothers. None of the other reports demonstrated a difference between insulin-treated and diet-treated newborns. The authors in the aforementioned studies explain the lack of difference in treated versus untreated women by the wide range in treatment initiation, from preconception to 38 weeks of gestation, and the considerable variability in insulin doses. Overall, data are lacking on the timing of insulin initiation, the criteria leading to insulin initiation, insulin dosage, oral hypoglycemic use, and maternal glycemic control. 

### 3.2. Influence of Insulin Treatment versus Diet on Gestational Age at Delivery 

Table 3 describes the influence of insulin treatment versus diet on gestational age at delivery in GCK(−) versus GCK(+) offspring. Two out of five studies in the GCK(−) group [13,17] and two out of five in the GCK(+) group [2,17] demonstrated a significantly earlier gestational age at delivery in the insulin-treated offspring, although they were at term. Others did not show a significant difference in gestational age at delivery. It may be that insulin-treated offspring were delivered earlier, as guidelines indicate at the discretion of the treating physician. We discuss preterm deliveries in a different section. 

### 3.3. Incidence of LGA, SGA, and Macrosomia 

The available data on newborn weight, including macrosomia, LGA, and SGA, are summarized in Table 4. As seen, out of 102 GCK(−) offspring, 44% were born LGA and 20% were born with macrosomia. Among the 173 GCK(+) offspring, 10% were LGA; 5% were SGA, which can be expected according to the normal distribution. The difference between the number of macrosomic and LGA newborns in the GCK(−) group compared to the GCK(+) group was statistically significant (*p* < 0.001) for both birthweight categories. 

## 4. Abortions

Three cohorts describe the abortion rate in the studied population. 

In Bacon et al., concerns were raised about the effect of maternal hyperglycemia on the miscarriage rate, which was 33% in GCK−MODY mothers compared to 15% in the background population. The miscarriages occurred at a median of 7.5 (6.2–8.7) weeks of gestation. The authors stated that the lack of available clinical studies necessitates implementing guidelines used for the treatment of GDM [14].

Dickens et al. [2] and Tinoco et al. [13] reported miscarriage rates of 19% and 17%, respectively, comparable to the background population rates. Both studies reported similar average gestational ages at the time of miscarriage, at approximately eight weeks of gestation. 

In our experience of two women and seven pregnancies, one pregnancy resulted in a spontaneous abortion at eight weeks.

Taken together, GCK−MODY women experience miscarriage rates similar to the general population, although given the available data, we cannot adequately account for the ’treatment’s contribution to this outcome. 

### 4.1. Congenital Anomalies in Newborns

Whether the rate of congenital anomalies in GCK−MODY pregnancies is increased is unknown. It has been reported that the incidence of congenital anomalies increases linearly with increasing HbA1C [7]. Recently, a case of sacral agenesis was reported in a fetus of a woman with GCK−MODY [7].

Congenital anomalies resulting in live births included a neural tube defect in an untreated GCK(−) offspring. [14], patent ductus arteriosus in an insulin-treated GCK(+) offspring [10], and four fetuses with general congenital malformations without data on mutational status or treatment [17]. Dickens et al. [2] reported four congenital anomalies resulting in miscarriages or elective termination of pregnancy. In our experience, one pregnancy was terminated due to microcephaly (with a normal karyotype) on the 17th week. 

Overall, data are scarce, and the possibility of congenital anomaly risk in women with GCK−MODY should be considered according to accepted parameters, including glycemic control [7]. 

### 4.2. Preterm Births (Before 37 Completed Weeks of Gestation)

We identified only two cohorts describing data on preterm births [2,18]. In the study by Dickens et al., the preterm birth rate was 12% [2]. Bitterman et al. [18] reported that 2 out of 20 GCK(+) (10%) offspring were born preterm. Neither of the studies reported the gestational age at delivery, the birthweight or centile, or information on treatment. Moreover, it was not reported whether the preterm deliveries were spontaneous or induced. One case report by Spyer et al. [19] reported a GCK(+) newborn treated with insulin was born at 36 weeks of gestation weighing 1610 g. Labor was induced due to fetal growth being consistently below the 10th percentile. 

### 4.3. Cesarean Deliveries 

Five cohorts describe data on CD rates, as seen in Table 5.

Two studies reported CD rates without differentiation for fetal mutational status. In Spyer et al. [17], 21/82 newborns (26%) were delivered by CD with a higher incidence of CD in the insulin-treated women compared with diet only (44% vs. 15%). In Bacon et al. [13], 19/41 live births (46%) were delivered by CD, and there was a higher incidence among insulin-treated women compared with diet only (57% vs. 43%). 

The other three studies made a distinction according to fetal mutational status. Dickens et al. [2] reported 2/23 (9%) GCK(+) and 3/12 (25%) GCK(−) offspring were delivered by CD. The numbers of insulin-treated versus diet-treated newborns are too small to consider statistical significance. In Tinoco et al. [14], 36% of deliveries were by CD, 28% in GCK(+) and 74% in GCK(−) newborns (*p* = 0.001). Insulin treatment in GCK(+) offspring was associated with an increased incidence of CD.

Collectively, and not surprisingly, the data suggest that insulin treatment influences the mode of delivery and leads to more interventions. However, most studies do not provide data on the indication for the CD (i.e., whether indicated due to a large baby or for another reason). There are no sufficient data to compare the significance between GCK(+) and GCK(−) newborns at this time. 

### 4.4. Shoulder Dystocia 

Three studies report shoulder dystocia as a neonatal complication (Table 5). Almost all cases were detected in GCK(−) offspring. The number of reported cases is too small to make a distinction between insulin-treated and untreated offspring. Tinoco et al. [14] noted three cases of shoulder dystocia, but there is no additional information about the group in which these complications occurred.

### 4.5. Long-Term Outcomes in Offspring

Several authors have raised concerns regarding the long-term outcomes of babies born to GCK−MODY mothers; however, little is known about these consequences and whether treatment will affect them. 

Two studies discuss the long-term impact on offspring born to GCK−MODY mothers and the effect of in utero exposure to hyperglycemia [12,16]. The first, by Singh et al., examined 86 adult offspring at a mean age of 40 years old [16]. Forty-nine were born to GCK mothers and exposed to hyperglycemia in utero, and thirty-seven were born to GCK fathers and served as the control group. Among the 29 GCK(−) offspring, 15 were born to an affected mother and 14 were born to an affected father. There are no details regarding treatment during pregnancy. Both groups had similar baseline characteristics (age, prediabetes and diabetes rates; BMI, and body fat percentage), and there were no significant differences in beta-cell function or glucose tolerance in GCK(−) offspring compared to the control group.

In a second study, by Fu et al., clinical and biochemical parameters were collected from 76 Chinese GCK(+) patients [12]. The mean age was 32 years, and 42% were males with a mean HbA1C of 6.5%. They found negative correlations between lower birthweight and 2 h postprandial glucose, glycated hemoglobin, total cholesterol (TC), and low-density lipoprotein cholesterol (LDL) after adjustment for age, gender, and BMI. Notably, the authors reported lower birthweights and higher levels of TC and LDL in seven GCK(+) adults whose mothers received insulin during pregnancy compared with adults born to mothers not treated with insulin, as depicted in Table 2. They suggest that a birthweight below 3100 g increases the risk for metabolic abnormalities, particularly dyslipidemia. This observation merits further consideration in other ethnic populations. 

### 4.6. Other Complications

There are few descriptions of neonatal hyperbilirubinemia, hypoglycemia, and respiratory distress, and, therefore, inferring a correlation to GCK−MODY is challenging. 

## 5. Discussions

The current treatment strategies for GCK−MODY pregnancies include those that suggest treating all women similarly to other forms of pre-gestational diabetes versus those favoring treatment based on the fetal mutational status. The recent literature supports the latter strategy, given data showing that untreated GCK(+) newborns are 550–700 g smaller than GCK(−) newborns and that using glucose-lowering treatment may further increase the risk of an SGA baby [9,15,17]. It is also noted that the required insulin dose can be higher than expected to reach glycemic control and may increase hypoglycemia risk. 

Our literature review identified nine retrospective cohort studies and five case reports exploring the impact of fetal mutational status on maternal and fetal outcomes. Based on these data, knowing the fetal mutational status is of great value due to the effect GCK−MODY has on the birthweight. Birthweight is significantly higher in GCK(−) offspring, with significantly higher rates of LGA and macrosomic newborns. Almost half of the reported GCK(−) newborns were born LGA and a fifth were macrosomic. The SGA percentage among GCK(+) newborns was 5%, which can be expected. Other complications, such as CD and shoulder dystocia, were more prevalent in the GCK(−) offspring. 

There are some data on abortions, congenital anomalies, preterm births, and other neonatal complications. However, the numbers are too small to reach any conclusions, and any outcome cannot be attributed to the mutational status, treatment, or the lack of it [23].

Data on whether to treat with insulin or treat with diet and follow mainly rely on reports of the birthweight and gestational age at delivery. Most available data show no difference with regard to insulin treatment versus diet only in the GCK(−) offspring, while some but not all reports found significantly smaller birthweights in the GCK(+) newborns. Possible explanations regarding the lack of difference in birthweight and gestational age at delivery are: 1. the small populations studied; 2. the retrospective analysis of chart data; 3. in some studies, treatment was self-reported and possibly biased or omitted; and 4. insulin was started too late or in insufficient doses. 

Data on insulin initiation, dose, timing, and glycemic control in the mother are mostly lacking; thus, variability in treatment makes drawing conclusions difficult. Inconsistent data correlate insulin treatment and gestational age at delivery in GCK(+) and GCK(−) pregnancies. Several reports found a positive association between insulin treatment and earlier gestational age at delivery, but all babies in those studies were born at term [2,13,17]. Hence, the effect of gestational age on the neonatal outcome is negligible. As for preterm deliveries, two studies reported rates of 10 to 12% in their MODY cohorts, which are comparable to the worldwide rate of preterm deliveries.

Long-term follow-up in GCK mutation carriers concerning the mode of treatment during pregnancy and later metabolic abnormalities is lacking. One study from China [12] on a very small number of cases showed that insulin treatment in GCK(+) newborns led to significantly lower weight babies with significantly higher rates of dyslipidemia at adulthood. There are no data on the actual cardiovascular risk or other ethnic groups. 

How can we incorporate the available literature into clinical decisions? A few points merit further consideration. 

In most pregnancies, the fetal mutational status remains unknown. If an invasive procedure is planned due to other indications, waiting for the fetal mutational status results until 18 to 20 weeks of gestation may delay the initiation of treatment of women with a GCK(−) fetuses, exposing both to complications related to fetal overgrowth. If an invasive procedure is not planned and fetal mutational status remains unknown, waiting until accelerated fetal growth appears (AC > 75th percentile) may also potentially lead to LGA or macrosomia and, as a result, to more interventions. Recent advances in noninvasive fetal genotyping using cell-free fetal DNA from maternal plasma sampling are promising [7,24].

Furthermore, a strategy of serial fetal AC measurements to guide insulin treatment initiation is subject to inherent measurement errors, which may be even more substantial with increasing pre-pregnancy overweight and obesity. Excessive gestational weight gain is also independently associated with accelerated fetal AC in late pregnancy, further complicating management decisions [7]. 

Although, outside of pregnancy, it is argued that the phenotype in GCK−MODY is similar regardless of genotype, this might not be the case during pregnancy. A study from Italy analyzed 20 GCK(+) children whose mothers did not receive treatment during pregnancy [18]. The average birthweight was 3130 g (range 2900–3500 g), and there were two SGA and three LGA neonates. All LGA newborns carried the same GCK mutation, and none of the other children had this mutation, suggesting that different GCK mutations may result in different phenotypes. Thus, even if data on fetal mutational status were known for all newborns, data are lacking on genotype–phenotype correlations. Structured models stratify the severity of impairment in the glucokinase gene into three categories: (1) drastic effect on catalytic activity; (2) reduced enzyme activity; and (3) a reduction of activity altering the interaction of glucokinase with other proteins such as the glucokinase regulatory protein. This genotypic variability may lead to a phenotypic difference that may be significant during pregnancy [25]. 

The genotype differences are manifested by a wide range in fasting glucose concentrations of 99–144 mg/dL and HbA1C of 5.7–7.5%. The assumption is that pregnancy outcomes do not depend on maternal baseline glucose levels. However, a baseline HbA1C of 7.5% might expose a woman to different pregnancy outcomes (congenital anomalies, abortion risk, and maternal complications) compared to a woman with an HbA1C of 5.7% at the beginning of her pregnancy. This potential difference might merit further consideration regarding insulin initiation while counseling a GCK−MODY woman before pregnancy and at the first prenatal visit. The reported wide range in blood glucose and insulin doses suggests that individual patient factors, such as co-existing insulin resistance, can influence the preferred treatment strategy during pregnancy. 

Additional unanswered issues relate to the glucose threshold for insulin initiation and, if initiated, what the appropriate glucose targets should be (i.e., similar to pre-gestational diabetes or “softer” targets). Further studies are needed to elucidate all of the above-mentioned unanswered issues.

## 6. Conclusions

To conclude, knowing the fetal mutational status is of great value. However, data show that solely knowing the fetal mutational status may not be sufficient to guide management, and other variables that complicate treatment decisions should be taken into consideration. After cautious interpretation of the limited available data, combined with our experience in managing these women during pregnancy, early insulin treatment should be considered in all GCK−MODY women, with a discontinuation of the treatment according to fetal mutational status if this becomes available later in pregnancy. Further data are needed on whether this will decrease the rate of LGA and macrosomic neonates and its potentially harmful consequences without significantly increasing the SGA rates. Additional research is essential to guide the management of pregnancies in women with GCK−MODY.

## Figures and Tables

**Table 1 ijerph-19-05980-t001:** Publications of Pregnant GCK−MODY Women with Known Fetal Mutational Status: Review of Maternal and Fetal Outcomes.

Retrospective Cohorts	Total Women	Total Pregnancies	Total GCK(+) ^†^	GCK(+) with Insulin	Total GCK(−) ^‡^	GCK(−) with Insulin
Hattersley AT 1998 [15]			19		21	
Singh R 2007 [16]			31		15	
Spyer 2009 [17]	42	82	44	14	38	19
Bacon 2015 [14]	12	56 (41 live births)	13	3	10	3
Bitterman 2018 [18]			20	0		
Dickens 2019 [2]	54	128	23	8	12	9
Hosokawa 2019 [10]	23	40	28	9	12	4
Fu 2019 [12]			28	7		
Tinoco 2021 [13]	34	119 (99 live births)	39	11	23	11
Case reports						
Spyer 2001 [19]	1	2	1	1	1	1
Chakera 2012 [20]	2	4	2			
Murphy 2015 [8]	1	1	1	1		
Haladova 2015 [21]	1	2			2	2
Udler 2020 [4]	1	4	1	1	3	3
Kirzhner 2022	2	7	1	1	1	1

^†^ GCK−affected offspring (GCK+); ^‡^ GCK−unaffected offspring (GCK−).

**Table 2 ijerph-19-05980-t002:** Birthweight and Birth Centiles in Treated Versus Untreated Pregnancies.

Retrospective Cohorts	GCK(−) Offspring Weight (kg) ^†^/Birth Centiles ^$^		*p* Value	GCK(+) Offspring Weight (kg) ^‡^/Birth Centiles ^$^		*p* Value
	No Insulin	Insulin		No Insulin	Insulin	
Spyer 2009 [17]	4.0 ± 0.5/86 ± 22	3.8 ± 0.6/84 ± 21	N/D ^¶^	3.3 ± 0.7/51 ± 30	3.0 ± 0.9/39.3 ± 33	N/D
Bacon 2015 [14]	4.1 (3.3–4.9)/88 ^$^	4 (3.8–4.1)/96 ^$^	N/A ^¶¶^	3.2 (3.1–3.7)/17 ^$^	3.3 (3–3.9)/40 ^$^	N/A
Dickens 2019 [2]	4.023/90 ± 8	3.757/84 ± 22	0.489/0.53	3.725/58 ± 33	2.967/34 ± 27	0.005 */0.11
Hosokawa 2019 [10]	3.593 ± 527/86 ± 10.3	3.025 ± 462/52.7 ± 37.1	0.154/0.048 *	2.800 ± 420/41.1 ± 31	2.532 ± 758/39.5 ± 31.3	0.595/0.885
Fu 2019 [12]				3.37 ± 0.39	2.83 ± 0.39	0.003 *
Tinoco 2021 [13]	3.9 ± 0.8/68.6 ± 33.9	4.2 ± 0.5/91.7 ± 17.8	0.42/0.06	3.4 ± 0.4/50.4 ± 28.1	3.5 ± 0.4/63.8 ± 35.1	0.54/0.22

^†^ GCK−unaffected offspring (GCK−); ^‡^ GCK−affected offspring (GCK+); ^¶^ N/D, no difference; ^¶¶^ N/A, not applicable; ^$^ Birthweight centiles according to available data. If data were not available, we estimated values according to the provided data of birthweight and GAD with the WHO fetal growth calculator; * statistically significant *p* value < 0.05.

**Table 3 ijerph-19-05980-t003:** Gestational Age at Delivery in Treated Versus Untreated Pregnancies.

Retrospective Cohorts	GCK(−) Offspring Gestational Age at Delivery ^†^		*p* Value	GCK(+) Offspring Gestational Age at Delivery ^‡^		*p* Value
	No Insulin	Insulin		No Insulin	Insulin	
Spyer 2009 [17]	38.9 ± 1.7	37.3 ± 1.1	<0.001 *	39.1 ± 2.7	37.8 ± 2.0	<0.05 *
Bacon 2015 [14]	40 (38–40)	38 (38–40)	N/A ^¶^	40 (39–40)	39 (38–40)	N/A
Dickens 2019 [2]	36.0	37.0	0.459	40.4	38.0	0.003 *
Hosokawa 2019 [10]	39.3	39.4	0.933	38.8	37.6	0.308
Tinoco 2021 [13]	39.5 ± 1.5	38.3 ± 1.0	0.03 *	39.6 ± 1.05	38.7 ± 1.4	0.07

^†^ GCK−unaffected offspring (GCK−); ^‡^ GCK−affected offspring (GCK+); ^¶^ N/A, not applicable; * statistically significant *p* value < 0.05.

**Table 4 ijerph-19-05980-t004:** Data on Birthweight.

Study (Year)		GCK(−) Offspring (*n* = 102)	GCK(+) Offspring (*n* = 173)
Spyer 2001 [19]	N (%)	1	1
	Macrosomia ^†^	0 (0)	0 (0)
	LGA ^‡^	0 (0)	0 (0)
	SGA ^¶^	0 (0)	1 (100)
Spyer 2009 [17]	N (%)	38	44
	Macrosomia	15 (39)	3 (7)
	LGA	21 (55)	4 (9)
	SGA	0 (0)	0 (0)
Chakera A 2012 [20]	N (%)	0	2
Murphy 2015 [8]	N (%)	0	1
Bacon 2015 [14]	N (%)	10	13
	Macrosomia	5 (50)	0 (0)
	LGA	NR ^¶¶^	NR
	SGA	0 (0)	1 (8)
Haladova 2015 [21]	N (%)	2	0
	Macrosomia	0 (0)	0 (0)
	LGA	1 (50)	0 (0)
	SGA	0 (0)	0 (0)
Bitterman 2018 [18]	N (%)	0	20
	Macrosomia	0 (0)	0 (0)
	LGA	0 (0)	3 (15)
	SGA	0 (0)	2 (10)
Dickens 2019 [2]	N (%)	12	23
	Macrosomia	N/A ^$^	N/A
	LGA	6 (50)	5 (38)
	SGA	0 (0)	0 (0)
Hosokawa 2019 [10]	N (%)	12	28
	Macrosomia	2 (16)	0 (0)
	LGA	N/A	N/A
	SGA	0 (0)	1 (4)
Udler 2020 [4]	N (%)	3	1
	Macrosomia	0 (0)	0 (0)
	LGA	1 (33)	0 (0)
	SGA	0 (0)	0 (0)
Tinoco 2021 [13]	N (%)	23	39
	Macrosomia	N/A	N/A
	LGA	15 (65)	5 (13)
	SGA	1 (4)	4 (10)
Kirzhner 2022	N (%)	1	1
	Macrosomia	0 (0)	0 (0)
	LGA	1 (100)	0 (0)
	SGA	0 (0)	0 (0)
Total N (%)	Macrosomia	22 (22)	3 (2)
	LGA	45 (44)	17 (10)
	SGA	1 (1)	9 (5)

^†^ Macrosomia is defined as newborns born ≥4 kg; ^‡^ Large for gestational age (LGA) is defined as newborns born ≥90th percentile; ^¶^ Small for gestational age (SGA) is defined as newborns born ≤ 10th percentile; ^¶¶^ NR, not recorded; ^$^ N/A, not applicable.

**Table 5 ijerph-19-05980-t005:** Available Data on Pregnancy Outcomes.

Retrospective Cohorts		GCK(−) Offspring		GCK(+) Offspring	
		No Insulin	Insulin	No Insulin	Insulin
Spyer 2009 [17]	Shoulder dystocia	4 (11)		0	
	C-section	21 (26) ^†^			
Bacon 2015 [14]	Shoulder dystocia	1	0	0	0
	C-section	19 (46) ^‡^			
	Complication	Neonatal hypoglycemia-1	0	1 *	0
Dickens 2019 [2]	Shoulder dystocia	0	0	1	0
	C-section	1 (one planned due to large baby)	2 (one emergency and one planned due to large baby)	2 (one emergency and one planned due to large baby)	0
	Complication	0	Respiratory issues-1	Prolonged labor due to large baby-2,meconium aspiration-1	Respiratory issues-2, hypoglycemia-1
Hosokawa 2019 [10]	Complication	0	0	0	Hyperbilirubinemia-2
Tinoco 2021 [13]	C-section	8 (67)	9 (82)	3 (11)	8 (73)
	Complication	Neonatal hypoglycemia-2	Neonatal hypoglycemia-4	0	Neonatal hypoglycemia-2
	Shoulder dystocia-3, fetal distress-1, hyperbilirubinemia-5				
Case reports					
Haladova 2015 [21]	C-section		2 (one due to cephalopelvic disproportion and imminent fetal hypoxia. Second unknown)		
	Complication		Hyperbilirubinemia-1, Neonatal hypoglycemia-1		
Udler 2020 [4]	Complication		Neonatal hypoglycemia-3		

^†^ incidence of C-section insulin-treated vs. noninsulin-treated, 44% vs. 15%; ^‡^ incidence of C-section insulin-treated vs. noninsulin-treated, 57.2% vs. 37.5%; * missing data.

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
