# Peer review of "The Challenges of Treating Glucokinase MODY during Pregnancy: A Review of Maternal and Fetal Outcomes"

_ijerph, 2022, doi:10.3390/ijerph19105980_

Round 1

Reviewer 1 Report

I have read with great interest the review by Kirzhner et al. Treating GCK MODY mothers during pregnancy is a great challenge and since data is scarce it is of utmost importance to expand it. A vigorous workup was made in this review, and although much remains to be studied, it was clearly shown that GCK(-) neonates suffer LGA and macrosomia and that insulin treatment is related more often to C-sections, unrelated to fetal mutation status.

Few minor remarks:

  1. Abstract:

The result section should include the total number of pregnancies studied (only GCK+ no. mentioned).

  1. Introduction:

In the 3rd criteria for genetic diagnosis, there appears to be a mistake - there is a usually only a small increment in glucose after OGTT

  1. Discussion:

*The main significant result found in the review is the increased rate of LGA and macrosomia in GCK(-) neonates. While SGA was the same for GCK(+) neonates. Hence, the main conclusion should be that knowing the fetus’s mutation status is of great value. All other pregnancy-related complications could not be related to GCK MODY. This should be emphasized more clearly in the discussion and conclusion sections.

*Apart from limited data and small retrospective samples, another limitation of the review should include that some of the data and case reports go back 20 years, when treatment and diagnostic modalities were different.

 - Spelling and typo mistakes should be revised (for example  Title- “and” should appear before Schiller Tal, p2 l15 controlled trials, p8l25 or induced)

Reviewer 2 Report

Dear authors,

I've appreciated a lot your paper

during the last days I'm facing to a case of query Mody pregnant woman therefore your literature review is crucially important for me and I'm sure of every obstetrician that has a special interest with gestational diabetes

I would only recommend you to focus, mention the role that chronic hyperglycemia may have on a short and long term risk

in particular for cases like mody where glicycaemia request major effort to be on target higher risk for maternal short and long term complication are present

I would like you to mention and cite this literature review:

doi: 10.1016/j.ajogmf.2021.100471. Epub 2021 Aug 25. PMID: 34454160.

in the discussion where you mention: 

...There are some data on abortions, congenital anomalies, preterm births, and other neonatal complications. However, numbers are too small to reach any conclusions. "

After this minor revision I would suggest it for publication
